# Bruton’s Tyrosine Kinase Inhibitors Ibrutinib and Acalabrutinib Counteract Anthracycline Resistance in Cancer Cells Expressing AKR1C3

**DOI:** 10.3390/cancers12123731

**Published:** 2020-12-11

**Authors:** Anselm Morell, Lucie Čermáková, Eva Novotná, Lenka Laštovičková, Melodie Haddad, Andrew Haddad, Ramon Portillo, Vladimír Wsól

**Affiliations:** 1Department of Biochemical Sciences, Faculty of Pharmacy, Charles University, Akademika Heyrovskeho 1203, 50005 Hradec Kralove, Czech Republic; morellga@faf.cuni.cz (A.M.); cermakoval@faf.cuni.cz (L.Č.); novotne7@faf.cuni.cz (E.N.); lastovile@faf.cuni.cz (L.L.); haddadme@faf.cuni.cz (M.H.); haddada@faf.cuni.cz (A.H.); 2Department of Pharmacology, Faculty of Pharmacy, Charles University, Akademika Heyrovskeho 1203, 50005 Hradec Kralove, Czech Republic; portillr@faf.cuni.cz

**Keywords:** AKR1C3, anthracyclines, Bruton’s tyrosine kinase, ibrutinib, acalabrutinib, multidrug resistance

## Abstract

**Simple Summary:**

The enzyme aldo-keto reductase 1C3 (AKR1C3) is present in several cancers, in which it is capable of actively metabolising different chemotherapy drugs and decreasing their cytotoxic effects. Therefore, the combination with specific inhibitors of AKR1C3 might prevent drug metabolism and increase its efficacy. We investigated the ability of Bruton’s tyrosine kinase inhibitors ibrutinib and acalabrutinib to block the AKR1C3 mediated inactivation of the anthracycline daunorubicin. Experimentation with recombinant AKR1C3 and different cancer cells expressing this enzyme outlined BTK-inhibitors as potential partners to synergise daunorubicin cytotoxicity in vitro. This evidence could be useful to improve the clinical outcome of anthracycline-based chemotherapies.

**Abstract:**

Over the last few years, aldo-keto reductase family 1 member C3 (AKR1C3) has been associated with the emergence of multidrug resistance (MDR), thereby hindering chemotherapy against cancer. In particular, impaired efficacy of the gold standards of induction therapy in acute myeloid leukaemia (AML) has been correlated with AKR1C3 expression, as this enzyme metabolises several drugs including anthracyclines. Therefore, the development of selective AKR1C3 inhibitors may help to overcome chemoresistance in clinical practice. In this regard, we demonstrated that Bruton’s tyrosine kinase (BTK) inhibitors ibrutinib and acalabrutinib efficiently prevented daunorubicin (Dau) inactivation mediated by AKR1C3 in both its recombinant form as well as during its overexpression in cancer cells. This revealed a synergistic effect of BTK inhibitors on Dau cytotoxicity in cancer cells expressing AKR1C3 both exogenously and endogenously, thus reverting anthracycline resistance in vitro. These findings suggest that BTK inhibitors have a novel off-target action, which can be exploited against leukaemia through combination regimens with standard chemotherapeutics like anthracyclines.

## 1. Introduction

Multidrug resistance (MDR) is one of the most challenging issues in cancer treatment. Cancer patients develop insensitivity to chemotherapeutic drugs, thereby increasing the chance for cancer recurrence and treatment failure. The following two main mechanisms are involved in impaired drug efficacy: the enhanced efflux through membrane transporters that reduce the drug’s intracellular concentration and its enzymatic transformation to less active metabolites [1,2,3]. A paradigm of MDR is the clinical use of anthracyclines [4,5], particularly daunorubicin (Dau). This class of drugs has been widely used over decades against solid tumours and haematological malignancies. However, its use is hindered by several reports of tumour resistance [6,7].

Two large superfamilies of NADPH-dependent carbonyl reducing enzymes (CREs), aldo-keto reductases (AKRs) and short-chain dehydrogenases/reductases (SDRs), represent some of the most important mediators in the inactivation of daunorubicin. Among them, CBR1, AKR1A1, 1B1, 1B10, 1C3, and 7A2 are the most active enzymes that impair Dau cytotoxicity through its reduction to the corresponding alcohol daunorubicinol (Dau-ol) [8,9]. Particularly, AKR1C3 has become a recurrent target in cancer, as its expression correlates with anthracycline resistance, and its function in prostaglandin and sex hormone biosynthesis may promote oncological disorders like leukaemia [10,11] and hormone-related tumours [12,13], respectively. Therefore, there is a need to characterise AKR1C3-specific inhibitors with clinical relevance in human disease [14]. In this regard, several AKR1C3 inhibitors have been reported to re-sensitise anthracycline-resistant cancer cells to daunorubicin both in vitro and ex vivo [15,16,17,18,19,20]. This scope aims to decipher novel drugs for combination regimens with standard chemotherapeutics like anthracyclines, which can counteract the MDR incidence.

In the present study, we outlined the effect of the Bruton’s tyrosine kinase (BTK) inhibitors ibrutinib (IBR) and acalabrutinib (ACA) on daunorubicin inactivation mediated by different CREs. BTK is a membrane protein belonging to the Tec-family of tyrosine kinases present in haematopoietic cells, except T or natural killer cells [21]. BTK has been mostly characterised in B cells, where it participates in B-cell antigen receptor (BCR)-mediated signalling pathways to trigger the phosphorylation of phospholipase C-γ2, which is essential for B cell activation, proliferation, and migration. Therefore, impaired BTK function has been associated with primary immunodeficiencies such as X-linked agammaglobulinemia (XLA) [22], whereas its upregulation underlies several B-cell malignancies, such as chronic lymphocytic leukaemia (CLL) or autoimmune diseases [23,24]. Based on this, ibrutinib was designed as a selective and irreversible inhibitor of BTK, thus blocking its signal transduction to prevent B-cell activation, with subsequent cell growth arrest and apoptosis. Therefore, ibrutinib improved the progression-free survival and overall survival in patients with CLL compared with conventional therapies [25,26,27], which enforced its approval by the US Food and Drug Administration (FDA) in 2013 for treatment of adult patients with mantle cell lymphoma or CLL. However, ibrutinib has shown inhibitory effects on other proteins, mostly kinases, not related to BCR signalling, thus producing off-target effects that develop treatment-limiting adverse events. Acalabrutinib was developed as a more potent, highly selective, covalent BTK inhibitor with lower off-target activity than ibrutinib and was approved by the US FDA in 2019. A previous study has shown that ibrutinib synergises daunorubicin cytotoxicity of AML cell lines through a potentially BTK-independent mechanism not related to its enhanced efflux [28], which remains to be elucidated. In this regard, our results showed that both IBR and ACA selectively inhibited AKR1C3-mediated inactivation of Dau in vitro and in cancer cell lines expressing AKR1C3, thus synergising Dau cytotoxicity to overcome anthracycline resistance.

## 2. Results

### 2.1. Acalabrutinib and Ibrutinib Inhibit the AKR1C3-Mediated Reduction of Daunorubicin In Vitro

First, we performed a screening of ACA and IBR effects on the different CREs involved in anthracycline metabolism [8]. To do this, we evaluated the metabolic activity of different recombinant CREs on Dau with two concentrations (10 and 50 µM) of BTK inhibitors (Figure 1).

Among all the enzymes analysed, only AKR1C3 underwent significant and equivalent inhibition by both BTK inhibitors (Figure 1D). The 10 and 50 µM concentrations of ACA and IBR reduced the activity of AKR1C3 by 89.55% ± 4%, 96.28% ± 1.58%, 92.17% ± 3.43%, and 96.41% ± 1.48%, respectively. Additionally, 10 and 50 µM IBR inhibited AKR1B10 activity (Figure 1C) significantly by 48.18% ± 2.10% and 74.97% ± 0.87%, respectively, whereas the highest concentration of ACA (50 µM) had a significant effect on AKR1B10 with a 9.36% ± 2.53% inhibition. We further analysed the type of interaction between both BTK inhibitors and AKR1C3. For this, the recombinant AKR1C3 was incubated with or without BTK inhibitors at three concentrations (0.5, 1, and 5 μM) in combination with increasing concentrations of Dau. The enzyme kinetics was evaluated with a Lineweaver-Burk double-reciprocal plot (Figure 2A), which showed a non-competitive/mixed phenotype for both compounds. This inhibition mode, together with a linear dependence of the IC_50_ value on the enzyme concentration (Figure 2B) suggested that these BTK inhibitors might display a tight-binding interaction with AKR1C3. Therefore, Morrison’s quadratic equation for tight-binding inhibitors [29] was applied with different amounts of AKR1C3 (Figure 2C) to calculate apparent Ki (Ki^app^) values (Table 1). Both compounds showed relatively constant Ki^app^ values in the nM range, with values lower for IBR than those for ACA.

### 2.2. BTK Inhibitors Counteract AKR1C3-Mediated Daunorubicin Resistance

To evaluate the effectiveness of IBR and ACA in counteracting the intracellular metabolism of Dau, we generated an HCT116 cell line transiently overexpressing the AKR1C3 enzyme (HCT116-C3) and control cells transfected with the empty vector (HCT116-EV). HCT116-C3 cells were treated with increasing concentrations of both BTK inhibitors (1, 5, and 10 µM) combined with Dau (1 µM) for 2.5 and 5 h. Both compounds inhibited the reduction of Dau to Dau-ol in a time- and dose-dependent manner (Figure 3A), demonstrating significant decreases in AKR1C3 activity of 29.40% ± 11.55%, 45.16% ± 0.86%, and 60.72% ± 6% with IBR; and 31.96% ± 13.51%, 67.15% ± 10.35%, and 65.56% ± 1.53% with ACA after 5 h. Furthermore, we analysed whether the decreased inactivation of Dau proportionally increased its cytotoxicity on cells. For this, Dau IC_50_ for HCT116-C3 and HCT116-EV were analysed in combination with the vehicle or same concentrations of BTK inhibitors (1, 5, and 10 µM) that previously showed an inhibitory effect on metabolism (Figure 3B,C and Table 2). The expression of AKR1C3 in HCT116-C3 cells conferred more resistance to Dau cytotoxicity, with the IC_50_ for Dau exhibiting a value that was approximately two times higher than that for HCT116-EV cells. However, Dau resistance in HCT116-C3 was reversed in a dose-dependent manner by IBR and ACA, with significant shifts in IC_50_ values for 5 and 10 µM of both BTK inhibitors. Nevertheless, in HCT116-EV cells, the combination with BTK inhibitors did not lead to an increase in Dau cytotoxicity, as no significant changes in Dau IC_50_ values were observed. Additionally, we analysed the combination pharmacodynamics of Dau and BTK inhibitors (Figure 3D and Appendix A). Using the Chou-Talalay method, the combination of Dau with 5 and 10 µM of both BTK inhibitors showed combination index (CI) values into the range between synergism (0.3–0.7) and strong synergism (0.1–0.3) for HCT116-C3 cells. However, for HCT116-EV cells, the CI values for the same combinations were much higher, demonstrating the range between moderate synergism (0.7–0.85) and near additivity (0.9–1.1). Likewise, in HCT116-C3 cells, the combination of both BTK inhibitors resulted in a dose-dependent reduction of the effective dose of Dau, whereas in HCT116-EV cells, this reduction was lower and unrelated to the concentrations of BTK inhibitors. Overall, our results showed that IBR and ACA could efficiently synergise Dau’s cytotoxicity on AKR1C3-expressing cells, thus overcoming Dau resistance by selectively inhibiting AKR1C3 activity. However, the mild synergism observed in HTC116-EV might be related to additional targets of BTK inhibitors but without sufficient contribution to significantly enhance Dau’s cytotoxicity.

### 2.3. BTK Inhibitors Synergise Daunorubicin Cytotoxicity in Cancer Cells

We have previously determined that IBR and ACA have a synergistic effect on Dau cytotoxicity related to the selective inhibition of AKR1C3 in transfected cells. Furthermore, similar experiments were performed with the lung epithelial cancer cell line A549, which constitutively express AKR1C3 [15]. First, we determined that both IBR and ACA (1, 5, and 10 µM) significantly reduced the intracellular metabolism of Dau in a dose-dependent manner in A549 cells (IBR: 15.20% ± 4.12%, 33.69% ± 2.74%, and 47.58% ± 6.01%; ACA: 12.13% ± 5.19%, 34.32% ± 7.09%, and 44.24% ± 9.06%) (Figure 4A). Likewise, we evaluated the cytotoxicity of Dau on this cell line in combination with the same concentrations of BTK inhibitors (Figure 4B and Table 3). Both BTK inhibitors displayed a certain degree of cytotoxicity which, despite being unrelated to Dau metabolism, may also contribute to the significant IC_50_ shifts observed after combining with Dau. However, similar to that observed previously with HCT116-AKR1C3, the improvement of the synergistic parameters evaluated by the Chou-Talalay method occurred in a dose-dependent manner of BTK inhibitors (Figure 4C and Appendix A). In conclusion, these results support the synergism between BTK inhibitors and Dau, which is partly related to the selective inhibition of Dau metabolism.

### 2.4. AKR1C3 Expression Is Not Correlated with the Effects of BTK Inhibitors

The lower activity of AKR1C3 may not be due to its selective inhibition but rather due to a reduction in its expression. Therefore, we analysed the effect of previously used concentrations of BTK inhibitors (1, 5, and 10 µM) on the protein levels of AKR1C3 in A549 cells after 72 h. Figure 5A,B shows the negligible or absence of effects of ACA and IBR on the expression of AKR1C3 in A549 cells. This implies that the Dau-sensitising effect observed in A549 cells is not related to the reduced expression of AKR1C3. On the other hand, previous studies have correlated the overexpression of AKR1C3 with reduced efficacy of protein kinase inhibitors in cancer cells. In this regard, we analysed the influence of AKR1C3 overexpression in HCT116 cells on the cytotoxicity of BTK inhibitors by determining the IC_50_ values (Figure 5C,D). For this, we calculated the relative resistance (RR) value by dividing the IC_50_ value of IBR or ACA among HCT116-C3 and HCT116-EV cells. Both cell lines showed equivalent sensitivities to IBR (RR = 1.06) and ACA (RR = 0.97). Hence, the overexpression of AKR1C3 did not affect the cytotoxic effect of BTK inhibitors.

### 2.5. Docking Prediction for the Binding of BTK Inhibitors with AKR1C3

To further understand the interaction between BTK inhibitors and AKR1C3 protein, we performed flexible docking with IBR and ACA on the crystal structure of AKR1C3 in complex with NADP+ (PDB: 1S2A [30]). The flexible model positioned both IBR and ACA into the catalytic site of AKR1C3 (Figure 6) with binding energies of −11.6 kcal/mol and −12.1 kcal/mol, respectively. The predicted pose for each BTK inhibitor was stabilised by several molecular interactions with AKR1C3 residues, mostly van der Waals and pi-interactions. IBR also showed conventional hydrogen bonding with the residues Ser-118, Trp-227, Arg-226, and Tyr-24, whereas ACA displayed stronger carbon-hydrogen bonding with Ser-129 and the cofactor NADP+. These interactions probably disrupted the conformation of residues and cofactor constituting the active site, thus forming an AKR1C3-inhibitor complex which could not reduce Dau despite binding to the active site. This may be in line with the non-competitive/mixed mode of inhibition observed between the IBR and ACA with AKR1C3.

## 3. Discussion

Apart from its association with various B-cell malignancies, BTK is also expressed in myeloid-lineage cells, showing constitutive activation in AML. In fact, BTK knockdown or inhibition by ibrutinib impaired the progression of AML in vitro, supporting BTK inhibitors as novel candidates in AML therapy. In this regard, different studies have analysed the synergistic effect of combining ibrutinib with gold standards of induction therapy in AML. Rushworth et al. [31] first reported that ibrutinib reduced the IC_50_ of daunorubicin and cytarabine by 3.5- and 1.5-fold, respectively, in primary AML samples. In the same study, ibrutinib-associated decrease in AML viability was correlated with BTK inhibition. However, BTK-independent targets were also suggested. Moreover, Zhang et al. [32] showed that ibrutinib inhibited the drug efflux function of multidrug resistance protein 1 (MRP1, ABCC1), thus enhancing the sensitivity of MRP1-overexpressing cells to those chemotherapeutic agents working as substrates for MRP1, such as the anthracyclines doxorubicin and daunorubicin. Later, Rotin et al. [28] observed that ibrutinib synergised daunorubicin on BTK-knockdown AML cells without increasing its accumulation, thus discarding BTK or MRP1 inhibition as the mechanism underlying this synergism. These findings led to the analysis of other ibrutinib targets that would explain synergism with daunorubicin, such as the SRC family kinases and other TEC family members [33]. For example, Griner et al. [34] suggested that ibrutinib’s inhibition of nuclear factor kappa B (NFκB) might counteract its protective function against anthracycline-induced DNA damage to explain the ibrutinib-doxorubicin synergism in B-cell lymphoma cells. A similar mechanism was observed in FLT3-ITD-negative AML cells, where ibrutinib suppressed NF-κB and STAT5-mediated transcription downstream of the TLR9/BTK transducer module [35].

In the present study, we deciphered the enzyme AKR1C3 as a novel BTK-independent target for ibrutinib and acalabrutinib. Among all the carbonyl reducing enzymes (CREs) participating in MDR, AKR1C3 reduces anthracyclines to less active C13-hydroxy products in a highly efficient manner [8]. Matsunaga et al. [36] demonstrated that the overexpression of AKR1C3 in the leukaemia cell line U937 confers resistance to chemotherapeutic drugs by enhancing daunorubicin metabolism. Herein, both BTK inhibitors could equally prevent the metabolic inactivation of daunorubicin by AKR1C3, which contributed to overcoming the inherent or acquired anthracycline resistance mediated by AKR1C3 expression in A549 and HCT116 cell lines. This is in line with previous reports from Verma et al. [16,17], showing the potential of several AKR1C3 inhibitors to strongly synergise the cytotoxic effect of daunorubicin (>10-fold) on AML cell lines expressing AKR1C3. We previously reported the inhibition of AKR1C3 as an off-target effect of the cyclin-dependent kinase (CDK) inhibitor dinaciclib [19], the phosphoinositide 3-kinase (PI3K) inhibitor buparlisib [37], and the FMS-like tyrosine kinase 3 receptor (FLT3) inhibitor midostaurin [38]. Dinaciclib and midostaurin, like ibrutinib [32], were described in parallel as specific inhibitors of the MRP1/ABCC1-mediated efflux of daunorubicin [39,40,41]. Expression of ABC transporter proteins in leukaemia cells has been thoroughly implicated in MDR [42]. However, Bailly et al. [43] argued that daunorubicin resistance in immature AML cells might not be exclusively related to drug efflux. Therefore, we showed that in the AML cell line KG1a, midostaurin synergised daunorubicin cytotoxicity by increasing its intracellular accumulation, probably by targeting the MRP1/ABCC1 transporter and by simultaneously inhibiting its metabolism by AKR1C3 [38]. This dual mechanism underlying the reversal of daunorubicin resistance in AML could be extended to ibrutinib and acalabrutinib in the case of targeting ABC transporter proteins as well. Based on our data, both BTK inhibitors inhibited AKR1C3 and consequently enhanced daunorubicin cytotoxicity in the concentration range of 1–10 μM, which was in concordance with ibrutinib concentrations (1–5 μM) that induced the chemosensitisation of MRP1-overexpressing cells [32]. However, this dual mechanism may conflict with the absence of daunorubicin accumulation reported in BTK-knockdown AML cells treated with similar ibrutinib concentrations of 4–8 μM [28]. Additionally, the effective concentrations of BTK inhibitors used in this study were higher than clinically achievable concentrations. For ibrutinib, a single oral dose of 140 mg resulted in a mean C_max_ concentration of 37.1 ng/mL [44], equivalent to ~84.2 nM, which extrapolated to the highest dosage recommended by the FDA (560 mg/day) with a maximum plasma concentration of ~0.34 μM. Nevertheless, this C_max_ is equivalent to the calculated concentrations (Ki^app^ values) required to achieve half-maximum inhibition of recombinant AKR1C3 which, together with the tight-binding interactions, ensures that AKR1C3 can be inhibited by ibrutinib at clinically relevant concentrations. In contrast, the recommended dose of acalabrutinib (200 mg/day) results in an approximate C_max_ of 825 ng/mL or 1.8 μM [45], which surpasses the minimum concentration required for synergism in our experiments with A549 cells. Therefore, we provide AKR1C3 as a specific target to explain the synergism between BTK inhibitors and daunorubicin in vitro. However, this may not be the only mechanism involved in the clinical setting.

Noteworthy, ibrutinib also significantly inhibited the AKR1B10-mediated reduction of daunorubicin in vitro. This may be relevant as Zhong et al. [46] have previously reported that AKR1B10 overexpression confers cell resistance to idarubicin and daunorubicin, whereas its specific inhibition significantly sensitises human lung cancer cells NCI-H460 to the same anthracyclines. In the present study, although lung adenocarcinoma cells A549 also expressed AKR1B10, this effect was not significant due to the following reasons: 10 μM of ibrutinib exhibited almost 2-fold the inhibitory potency over recombinant AKR1C3 than that observed with AKR1B10, even when AKR1B10 had a 4.7-fold lower specific activity compared to that of daunorubicin [38]. Further, despite the fact that 10 μM of acalabrutinib did not show significant inhibition of the recombinant AKR1B10, this concentration displayed equivalent reduction of daunorubicin metabolism and the related increase in A549 cytotoxicity with ibrutinib. However, AKR1B10 is involved in the positive regulation of cell proliferation; therefore, its overexpression correlates with the carcinogenesis of different tissues. Likewise, AKR1B10 inhibition or knockdown by small interfering RNA reduced the proliferation of different cancer cell lines [47,48,49]. AKR1B10 is also significantly overexpressed in several leukaemias [50]; therefore, the implications of its targeting by ibrutinib should be further evaluated. Similarly, the key functions of AKR1C3 in the metabolism of prostaglandins and sex hormones have been related to the development of leukaemia [10,11] and hormone-related tumours [12,13], respectively; thus, AKR1C3 targeting by BTK inhibitors may also counteract carcinogenesis in those tissues overexpressing AKR1C3. For this reason, as a future scope of research, we aim to study the mechanisms by which BTK inhibitors target AKR1C3 and AKR1B10 and modulate cell proliferation for potential applications in cancer treatment.

## 4. Materials and Methods

### 4.1. Chemicals

Ibrutinib and acalabrutinib were obtained from MCE (MedChemExpress, Monmouth Junction, NJ, USA). NADP+, glucose-6-phosphate, and HPLC-grade solvents were purchased from Sigma-Aldrich (Prague, Czech Republic). Daunorubicin and daunorubicinol were obtained from Toronto Research Chemicals (Toronto, ON, Canada). Glucose-6-phosphate dehydrogenase was provided by Roche Diagnostics (Mannheim, Germany). Cell culture reagents were provided by Lonza (Walkersville, MD, USA) and PAA Laboratories (Pashing, Austria).

### 4.2. Determination of Dau Metabolism by Recombinant CREs

Human recombinant CREs (CBR1, AKR1A1, 1B1, 1B10, 7A2, and 1C3) were produced by an Escherichia coli expression system as per methods described previously [19,51,52]. For the inhibitory assay, each recombinant enzyme, CBR1 (1 µg), AKR1A1 (1 µg), AKR1B1 (5 µg), AKR1B10 (5 µg), AKR7A2 (3 µg), or AKR1C3 (1.5 µg) was incubated with BTK inhibitors at different concentrations (10 and 50 µM) and the substrate Dau (500 µM). For analysis using the Lineweaver-Burk plot, 1.5 µg of recombinant AKR1C3 was incubated with DMSO or BTK inhibitors (1, 5, and 10 µM) and Dau (200, 400, 600, 800, 1000, and 2000 µM). To identify the tight-binding inhibition of AKR1C3, the IC_50_ values of BTK inhibitors (0.01, 0.1, 0.25, 0.5, 1, 5, 10, and 50 µM) were determined at different concentrations of AKR1C3 (1.35, 0.81, 0.40, and 0.13 µM) and fitted to the Morrison equation implemented in GraphPad Prism 8.1.2. (Km = 387.2 ± 36.01 µM) to quantify the inhibition constant Ki^app^. The incubations and further metabolite detection were further performed by using ultra-high-performance liquid chromatography (UHPLC) as per methods described previously [18].

### 4.3. Cell Cultures

HCT116 human colorectal carcinoma cell line was obtained from the European Collection of Authenticated Cell Cultures (Salisbury, UK) and A549 lung adenocarcinoma cells were obtained from the American Type Culture Collection (Manassas, VA, USA). Both cell lines were cultured in DMEM with 10% FBS at 37 °C in a 5% CO_2_ atmosphere in humidified air.

### 4.4. Transient Transfection of HCT116 Cells

HCT116 cells were seeded (1.25 × 10^5^ cells/well) in 24-well plates and cultured for 24 h. For the transient expression of the recombinant human AKR1C3, the cells were transfected with a pCI_AKR1C3 or empty pCI plasmid as control (previously described in a study by Hofman et al. [15]) according to the manufacturer’s instructions (jetPRIME^®^ Transfection Reagent; Polyplus Transfection^®^, Illkirch, France). For each 24-well plate, a mixture of 0.75 µL of the jetPRIME^®^ Transfection Reagent and 0.25 µg of plasmid was incubated for 10 min at room temperature. Then, polyplexes were added to the HCT116 cells with 0.5 mL of fresh DMEM supplemented with 10% FBS and incubated for 24 h (37 °C, 5% CO_2_). The expression of recombinant AKR1C3 was verified by western blotting as per methods described previously [38].

### 4.5. Determination of Dau Metabolism by Cells

After transfection of HCT116 cells or seeding of A549 cells (25 × 10^3^ cells/well) in 24-well plates for 24 h, the medium was harvested and replaced with DMEM supplemented with 10% FBS containing 1 µM daunorubicin, with or without increasing concentrations (1, 5, and 10 μM) of BTK inhibitors. Cells were incubated (37 °C, 5% CO_2_) for 2.5 and 5 h (transfected HCT116) or only 5 h (A549). Then, the medium was collected and the cells were lysed in lysis buffer (25 mM Tris, 150 mM NaCl, and 1% Triton X-100, pH 7.8) for 15 min at room temperature. The collected medium was combined with cell lysate for further extraction and analysis of metabolites, as per methods described previously [18].

### 4.6. Determination of Synergism

The synergistic effect of BTK inhibitors on Dau cytotoxicity was evaluated in transfected HCT116 and A549 cells by IC_50_ determination. Twenty-four hours prior to the experiment, transfected HCT116 cells (8 × 10^3^ cells/well) or A549 cells (5 × 10^3^ cells/well) were seeded in 96-well plates. The cell culture medium was replaced with fresh medium containing increasing amounts of Dau (final concentrations: 0.01, 0.05, 0.1, 0.25, 0.5, 0.75, 1, and 10 µM) with or without BTK inhibitors (final concentrations: 1, 5, and 10 µM). After 72 h of incubation under standard conditions (37 °C, 5% CO_2_), cell viability was assessed by MTT (Sigma-Aldrich, Prague, Czech Republic) assay. For this, 50 µL of the MTT solution (3 mg/mL) was added to each well and incubated for 30 min. The supernatant was collected, and the cell monolayers were dissolved in 100 µL of DMSO. The absorbance was measured at 570 and 690 nm by a microplate reader Infinite M200 (Tecan, Salzburg, Austria), and the background values at 690 nm were subtracted from the absorbance obtained at 570 nm. The half-maximal inhibitory concentrations (IC_50_) were calculated using GraphPad Prism 8.1.2. The drug combination pharmacodynamics were determined by the Chou-Talalay method using the CompuSyn ver. 3.0.1 software (ComboSyn Inc., Paramus, NJ, USA).

### 4.7. Flexible Docking of BTK Inhibitors in the Catalytic Site of AKR1C3

The AKR1C3 crystal structure in complex with NADP+ and indomethacin (PDB ID: 1S2A [30]) was downloaded from the RCSB protein database (rcsb.org) and prepared for docking using MGL Tools Utilities (MGL Tools 1.5.6) [53]. The water molecules were removed, polar hydrogens were added, and non-polar hydrogens were merged. Gasteiger charges were assigned to all atoms. Structures of acalabrutinib and ibrutinib were obtained from the ZINC Database [54] (http://zinc.docking.org). The ligands were prepared for docking with UCSF Chimera 1.14 with an Amber force field [55] and AutoDockTools. Flexible docking (flexible residues Tyr-24, Tyr-55, Trp-86, His-117, Tyr-216, Trp-227, Phe-306, and Phe-311) was performed with AutoDock Vina 1.1.2 [56] (coordinates 28.911 × −26.519 × 59.430; exhaustiveness parameter 8). A grid box with dimensions 23.127 × 23.127 × 23.127 for acalabrutinib and a grid box with dimensions 22.535 × 22.535 × 22.535 for ibrutinib were used for docking, as per methods reported by Feinstein & Brylinski [57]. AKR1C3 and ligand interactions were visualised using the BIOVIA Discovery Studio Visualiser by Dassault Systèmes (San Diego, CA, USA) [58].

### 4.8. Statistical Analysis

Significant changes in Dau metabolism and AKR1C3 protein expression were assessed by using one-way ANOVA followed by Dunnett’s post hoc test. The Student’s t-test was used to determine the statistical significance of the IC_50_ shift. Whole changes in the metabolism of Dau at different times were analysed by using two-way ANOVA followed by Bonferroni’s post hoc test. Statistical analysis was performed using GraphPad Prism 8.1.2. A *p* value < 0.05 was considered statistically significant.

## 5. Conclusions

In conclusion, this study outlined the ability of the BTK inhibitors ibrutinib and acalabrutinib to prevent the AKR1C3-mediated reduction of daunorubicin, which contributed to their synergistic effect on cancer cells expressing AKR1C3. These results support previous evidence encouraging the concomitant administration of BTK inhibitors to improve the therapeutic efficacy of anthracyclines and counteract multidrug resistance in AML.

## Abbreviations

ABCATP-binding cassetteAKRAldo-keto reductaseAMLAcute myeloid leukaemiaCICombination indexCRECarbonyl-reducing enzymeDauDaunorubicinDau-olDaunorubicinolDMSODimethyl sulfoxideFaFraction affectedFLT3FMS-like tyrosine kinase 3 receptorHCT116-AKR1C3Cells transfected with pCI_AKR1C3MidMidostaurinMTT3-(4,5-Dimethylthiazoyl-2-yl)2,5-diphenyl tetrazolium bromideMDRMultidrug resistanceNADPHNicotinamide adenine dinucleotide phosphatePGD2Prostaglandin D211β-PGF2α11β-Prostaglandin F2αPPARγPeroxisome proliferator-activated receptor gammaSDStandard deviationSDRShort-chain dehydrogenase/reductaseTKITyrosine kinase inhibitorUHPLCUltra high-performance liquid chromatographyWTWild type15dPGJ215-deoxyΔ^12,14^PGJ2

## Figures and Tables

**Figure 1 cancers-12-03731-f001:**
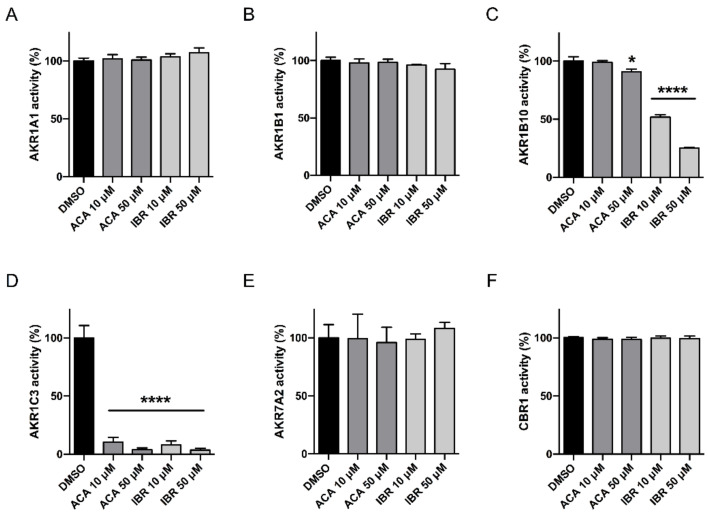
Screening of BTK inhibitors’ effect on different CRE-mediated reduction of Dau. Bars represent the in vitro activity over Dau of recombinant AKR1A1 (**A**), AKR1B1 (**B**), AKR1B10 (**C**), AKR1C3 (**D**), AKR7A2 (**E**), and CBR1 (**F**) in the presence of the cofactor NADPH. The enzymatic activity is expressed as the percentage relative to DMSO (*n* = 3; mean ± SD). The data were subjected to statistical analysis using one-way ANOVA followed by Dunnett’s post hoc test. * *p* < 0.01, **** *p* < 0.0001.

**Figure 2 cancers-12-03731-f002:**
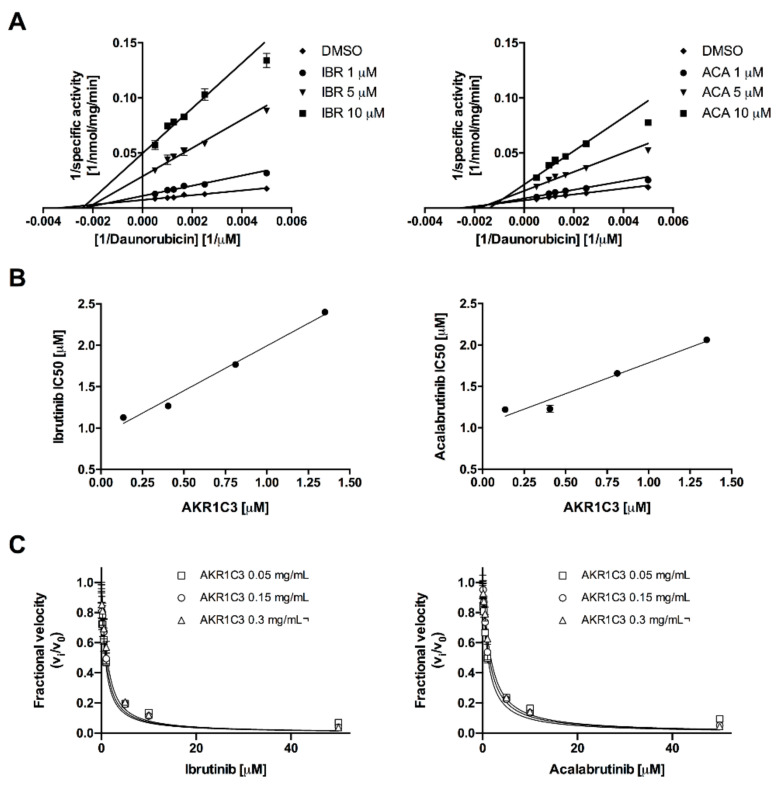
Analysis of steady-state kinetics of BTK inhibitors’ effect on AKR1C3-mediated reduction of Dau. (**A**) Lineweaver-Burk plot comparing the inverse AKR1C3-specific activity in the presence of DMSO or increasing concentrations (1, 5, and 10 µM) of BTK inhibitors as a function of the inverse of the Dau concentration (*n* = 3; mean ± SD). (**B**) Measured IC_50_ values of BTK inhibitors as a function of the AKR1C3 concentration (*n* = 3; mean ± SD). (**C**) Concentration-response plot of BTK inhibitors for AKR1C3-catalysed Dau activity fitted to Morrison’s quadratic equation. The enzymatic activity is expressed as the ratio of inhibited versus the non-inhibited reaction rate (*n* = 6; mean ± SD).

**Figure 3 cancers-12-03731-f003:**
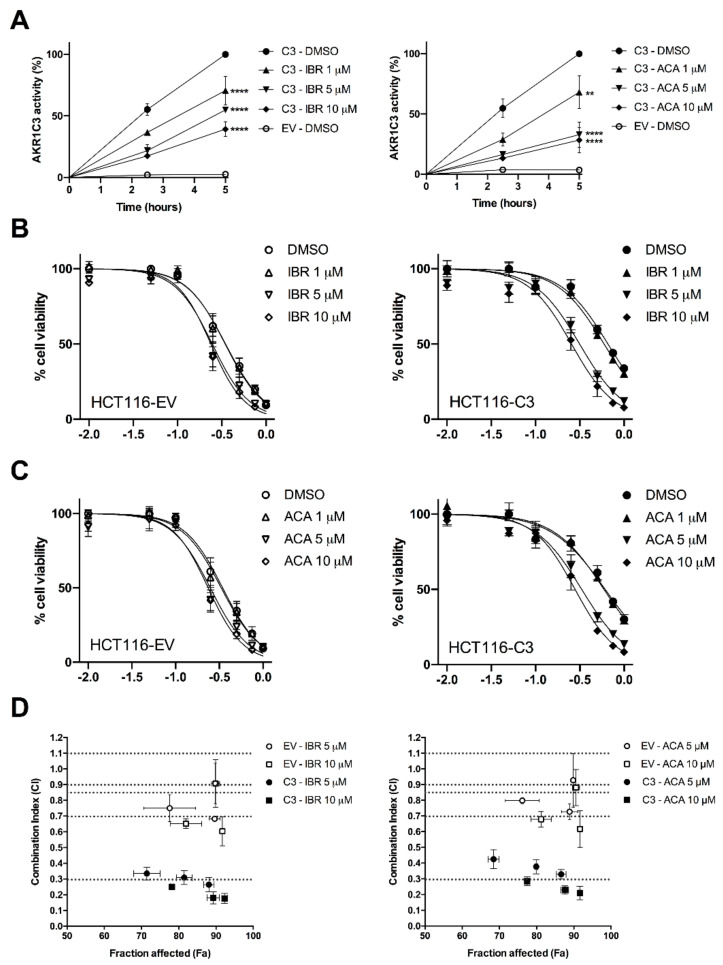
BTK inhibitors counteract AKR1C3-mediated Dau resistance. (**A**) Transfected HCT116 cells were incubated with 1 µM of Dau in combination with DMSO or BTK inhibitors (1, 5, and 10 µM) for 2.5 and 5 h, and the cell extracts were further analysed by UHPLC. The graph represents the time-dependent AKR1C3 activity, considering the amount of Dau-ol produced after 5 h in HCT116-C3 cells as 100% and that in HCT116-EV cells as 0%. All the values are expressed as mean ± SD of three independent experiments performed in duplicate (*n* = 6). The data were subjected to statistical analysis using two-way ANOVA followed by Bonferroni’s post hoc test. ** *p* < 0.01 and **** *p* < 0.0001 regarding DMSO. (**B**,**C**) HCT116-EV and HCT116-C3 were exposed to vehicle or different doses of BTK inhibitors (1, 5, and 10 µM) in combination with increasing concentrations of Dau and incubated for 72 h. The cell viability was assessed by MTT assay. Graphs represent a comparison of the dose-response curves for each cell line viability (%) under the different combinations (*n* = 6, mean ± SD). (**D**) Combination index (CI) vs. fraction-affected (Fa) plots obtained after the Chou-Talalay analysis of the combined treatment involving BTK inhibitors (5 and 10 µM) with Dau (0.1–1 µM) (*n* = 6, mean ± SD). Dashed lines delineate the CI ranges for strong synergism (0.1–0.3), synergism (0.3–0.7), moderate synergism (0.7–0.85), and near additivity (0.9–1.1).

**Figure 4 cancers-12-03731-f004:**
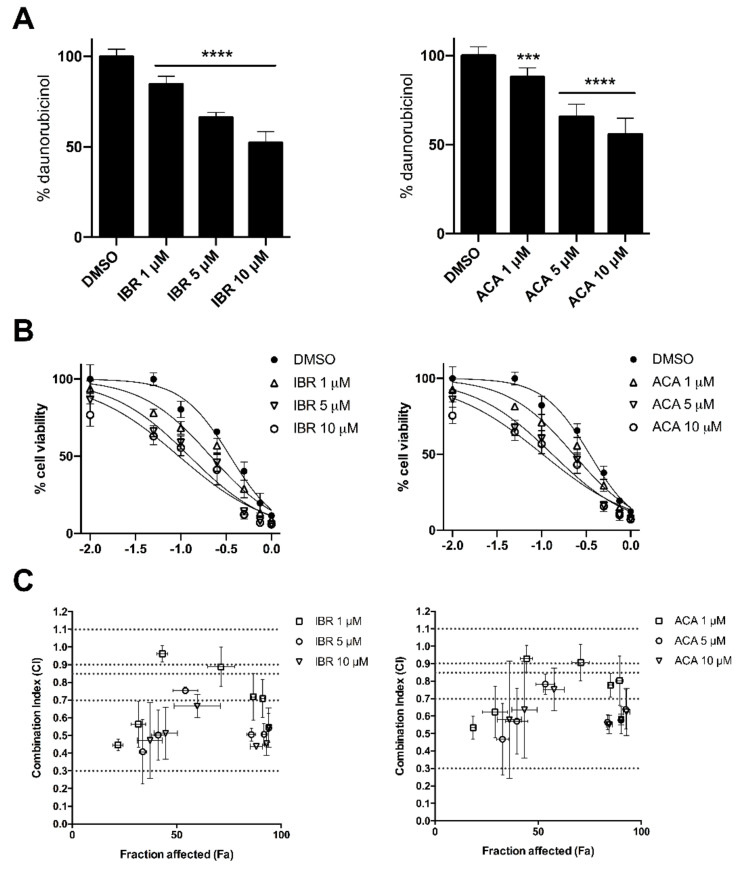
BTK inhibitors synergise daunorubicin cytotoxicity in cancer cells. (**A)** A549 cells were incubated with 1 µM of Dau in combination with DMSO or BTK inhibitors (1, 5, and 10 µM) for 5 h, and the cell extracts were further analysed by UHPLC. Bars represent daunorubicinol amounts relative to control in percentage. All the values are expressed as mean ± SD of three independent experiments performed in duplicate (*n* = 6). The data were subjected to statistical analysis using one-way ANOVA followed by Dunnett’s post hoc test. *** *p* < 0.001; **** *p* < 0.0001. (**B**) A549 cells were incubated with vehicle or non-toxic doses of BTK inhibitors (1, 5, and 10 µM) in combination with increasing concentrations of Dau for 72 h, followed by MTT assay to determine cell viability. Graphs represent a comparison of the dose-response curves of A549 cells viability (%) under the different combinations (*n* = 9, mean ± SD). (**C**) Combination index (CI) vs. fraction-affected (Fa) plots obtained after the Chou-Talalay analysis of the combined treatment involving BTK inhibitors (1, 5, and 10 µM) with Dau (0.1–1 µM) (*n* = 6, mean ± SD). Dashed lines delineate the CI ranges for strong synergism (0.1–0.3), synergism (0.3–0.7), moderate synergism (0.7–0.85), and near additivity (0.9–1.1).

**Figure 5 cancers-12-03731-f005:**
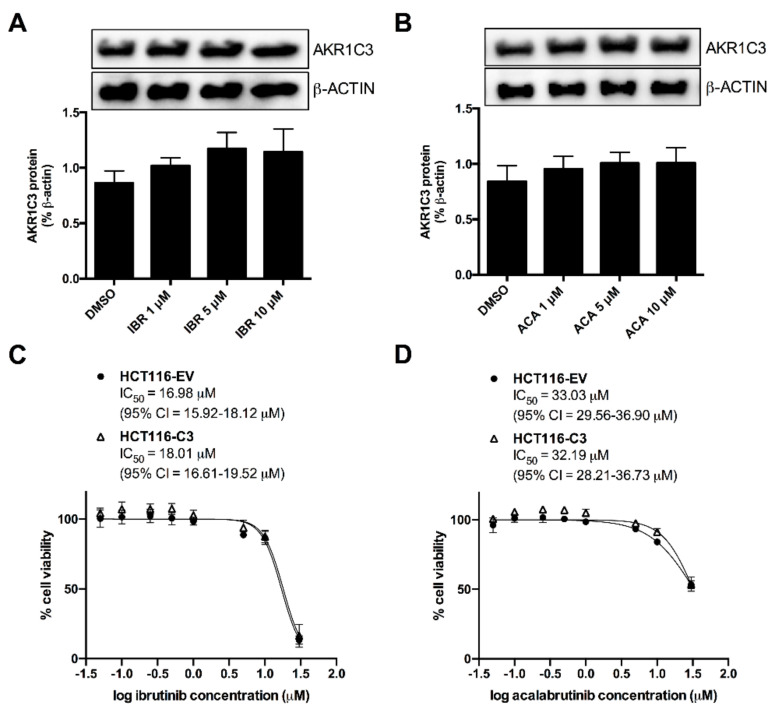
There is no correlation between AKR1C3 expression and effects of BTK inhibitors. A549 cells were incubated with DMSO or BTK inhibitors (1, 5, and 10 µM) for 72 h, and then total protein was extracted for analysis of AKR1C3 expression by western blotting (WB) as per methods described previously [20]. A representative WB is shown for IBR (**A**) and ACA (**B**) incubations, and bars represent AKR1C3 protein levels as determined by densitometry relative to β-actin (*n* = 3, mean ± SD). Uncropped images from WB are available in Appendix A. On the other hand, HCT116-EV and HCT116-C3 cells were treated with increasing concentrations of IBR (**C**) and ACA (**D**) for 72 h, and their viability was assessed by MTT assay. Graphs represent a comparison of the dose-response curves for percentages of cell viability in each cell line (*n* = 6, mean ± SD).

**Figure 6 cancers-12-03731-f006:**
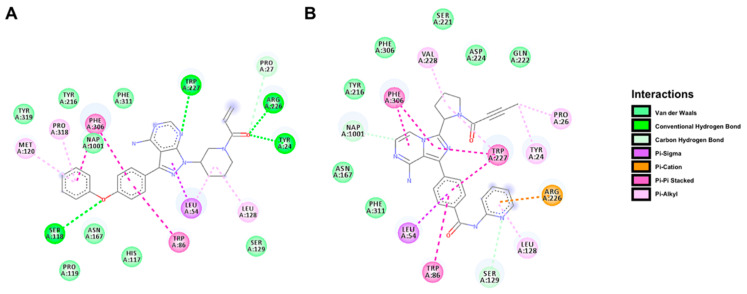
Docking prediction illustrating the binding between BTK inhibitors and AKR1C3. Images show the 2D view of the lowest energy pose of IBR (**A**) and ACA (**B**) within the AKR1C3 crystal structure (PDB: 1S2A). The predicted interactions between each compound and the surrounding amino acids and cofactor NADP+ (NAP) are depicted by different colours and listed on the right side.

**Table 1 cancers-12-03731-t001:** Ki^app^ values for BTK-inhibitors obtained from the Morrison equation for tight- binding inhibitors.

BTK-Inhibitor	AKR1C3 (mg/mL)	AKR1C3 (μM)	Ki^app^ (μM)	95% CI (μM)	R^2^
Ibrutinib	0.005	0.135	0.312	0.242–0.382	0.881
	0.015	0.405	0.332	0.268–0.396	0.944
	0.030	0.811	0.354	0.278–0.431	0.951
Acalabrutinib	0.005	0.135	0.451	0.356–0.546	0.946
	0.015	0.405	0.521	0.454–0.589	0.983
	0.030	0.811	0.566	0.496–0.636	0.987

Values are expressed as the means ± SD from at least three independent experiments.

**Table 2 cancers-12-03731-t002:** BTK-inhibitors revert the resistance to Dau mediated by AKR1C3 expression in HCT116 cells.

BTK-Inhibitor	Concentration (μM)	IC_50_ (95% CI) [μM]	
		HCT116-EV	HCT116-C3
Ibrutinib	0	0.35 (0.33–0.37)	0.66 (0.62–0.70)
	1	0.35 (0.32–0.37)	0.59 (0.56–0.63)
	5	0.25 (0.22–0.27)	0.31 (0.29–0.34) **
	10	0.24 (0.22–0.26)	0.25 (0.23–0.28) **
Acalabrutinib	0	0.34 (0.32–0.37)	0.61 (0.57–0.66)
	1	0.33 (0.31–0.36)	0.59 (0.55–0.64)
	5	0.25 (0.23–0.27)	0.33 (0.31–0.36) **
	10	0.23 (0.22–0.26)	0.28 (0.26–0.30) **

IC_50_ values are expressed as mean of two independent experiments performed in triplicate. The IC_50_ values were subjected to statistical analysis using the Student’s *t*-test. ** *p* < 0.001.

**Table 3 cancers-12-03731-t003:** BTK-inhibitors enhance the sensitivity of A549 cells to Dau cytotoxicity.

BTK-Inhibitor	Concentration (μM)	IC_50_ (95% CI) [μM]
Ibrutinib	0	0.35 (0.33–0.37)
	1	0.21 (0.19–0.23) **
	5	0.12 (0.11–0.14) ***
	10	0.09 (0.08–0.11) ***
Acalabrutinib	0	0.35 (0.33–0.37)
	1	0.23 (0.21–0.25) **
	5	0.13 (0.12–0.15) ***
	10	0.10 (0.09–0.12) ***

IC_50_ values are expressed as mean of three independent experiments performed in triplicate. The IC_50_ values were subjected to statistical analysis using the Student’s *t*-test. ** *p* < 0.001; *** *p* < 0.0001.

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
