# Peer review of "Bruton’s Tyrosine Kinase Inhibitors Ibrutinib and Acalabrutinib Counteract Anthracycline Resistance in Cancer Cells Expressing AKR1C3"

_cancers, 2020, doi:10.3390/cancers12123731_

Round 1
Reviewer 1 Report
The manuscript cancers-1023741, entitled “Bruton's tyrosine kinase inhibitors ibrutinib and acalabrutinib counteract anthracycline resistance in cancer cells expressing AKR1C3” by Morell and coworkers is certainly interesting and edge-cutting.
The manuscript is well structured and written. The results are supported with a number of adequate approaches. The research is timely and the hypothesis is well meditated.
With their experiments the authors open a new path to avoid multidrug resistance in cancer cells, a handicap in the treatment and eradication on several cancer types. The authors show that Bruton's tyrosine kinase inhibitors may regulate the function of AKR1C3, which metabolizes a number of classical anti-tumoral drugs employed in cancer treatment, anthracyclines among then. Treatment of cancer cells with brutinib and acalabrutinib resulted in the impairment of daunorubicin inactivation mediated by AKR1C3.
I would suggest a figure explaining the role of AKR1C3 in MTD in cancer cells, a sort of graphical abstract is you wish, to help the reader to further understand the ms.
Author Response
Response to Reviewer 1 Comments
Point 1: I would suggest a figure explaining the role of AKR1C3 in MTD in cancer cells, a sort of graphical abstract is you wish, to help the reader to further understand the ms.
Response 1: We are not sure if the Reviewer has seen the graphical abstract, which we submitted together with our manuscript, in which we tried to express the main idea of our manuscript.
Reviewer 2 Report
The study by Morell and co-workers presents evidence that the BTK inhibitor ibrutinib and acalabrutinib have off-target BTK-independent inhibitory activity against the drug-metabolizing enzyme aldo-keto reductase 1C3 (AKR1C3). By inhibiting AKR1C3 in vitro and in AKR1C3 expressing cells, both BTK inhibitors are able to block the drug metabolizing activity of AKR1C3 and enhance the toxicity of chemotherapeutic cancer drug daunorubicin. Even though this is to my knowledge the first report of such an activity for these compounds, the evidence is quite intriguing and convincing, worthy of further research and confirmation.
A couple of editorial errors need to be corrected, and acceptance is recommended.
1) Line 14: "where actively metabolizes" should be revised to be grammatically correct.
2) Lines 59 and 60: it is not clear what is meant by "its biosynthetic functions may promote oncological disorders".
3) Lines 79-81: The statement needs references.
Author Response
Response to Reviewer 2 Comments
Point 1: Line 14: "where actively metabolizes" should be revised to be grammatically correct.
Response 1: Except for the Simple Summary, the entire manuscript was linguistically edited in Elsevier (I enclose the certificate). We corrected this sentence to "The enzyme aldo-keto reductase 1C3 (AKR1C3) is present in several cancers, in which it is capable of actively metabolizing different chemotherapy drugs and decrease their cytotoxic effects."
Point 2: Lines 59 and 60: it is not clear what is meant by "its biosynthetic functions may promote oncological disorders".
Response 2: We corrected the sentence to "Particularly, AKR1C3 has become a recurrent target in cancer, as its expression correlates with anthracycline resistance, and its function in prostaglandin and sex hormone biosynthesis may promote oncological disorders like leukaemia [10,11] and hormone-related tumours [12,13], respectively.
Point 3: Lines 79-81: The statement needs references.
Response 3: We added 3 new references
[25] J.C. Byrd, R.R. Furman, S.E. Coutre, I.W. Flinn, J.A. Burger, K.A. Blum, B. Grant, J.P. Sharman, M. Coleman, W.G. Wierda, J.A. Jones, W. Zhao, N.A. Heerema, A.J. Johnson, J. Sukbuntherng, B.Y. Chang, F. Clow, E. Hedrick, J.J. Buggy, D.F. James, S. O’Brien, Ibrutinib versus ofatumumab in previously treated chronic lymphoid leukemia. N Engl J Med. 2014, 371(3), pp. 213-223. https://doi.org/10.1056/nejmoa1215637.
[26] J.C. Byrd, J.R. Brown, S. O’Brien, J.C. Barrientos, N.E. Kay, N.M. Reddy, S. Coutre, C.S. Tam, S.P. Mulligan, U. Jaeger, S. Devereux, P.M. Barr, R.R. Furman, T.J. Kipps, F. Cymbalista, C. Pocock, P. Thornton, F. Caligaris-Cappio, T. Robak, J. Delgado, S.J. Schuster, M. Montillo, A. Schuh, S. de Vos, D. Gill, A. Bloor, C. Dearden, C. Moreno, J.J. Jones, A.D. Chu, M. Fardis, J. McGreivy, F. Clow, D.F. James, P. Hillmen, Targeting BTK with ibrutinib in relapsed chronic lymphocytic leukemia. N Engl J Med. 2013, 369(1), pp. 32-42. https://doi.org/10.1056/nejmoa1400376.
[27] J.A. Burger, A. Tedeschi, P.M. Barr, T. Robak, C. Owen, P. Ghia, O. Bairey, P. Hillmen, N.L. Bartlett, J. Li, D. Simpson, S. Grosicki, S. Devereux, H. McCarthy, S. Coutre, H. Quach, G. Gaidano, Z. Maslyak, D.A. Stevens, A. Janssens, F. Offner, J. Mayer, M. O’Dwyer, A. Hellmann, A. Schuh, T. Siddiqi, A. Polliack, C.S. Tam, D. Suri, M. Cheng, F. Clow, L. Styles, D.F. James, T.J. Kipps, Ibrutinib as initial therapy for patients with chronic lymphocytic leukemia. N Engl J Med. 2015, 373(25), pp. 2425-2437. https://doi.org/10.1056/nejmoa1509388.